# Characterization of the Effect of the CO₂/CO Sintering Atmosphere on the Abrasion Resistance of a FeCrCB Hardfacing Coating

**Fernando Valenzuela-De la Rosa, Roal Torres-Sánchez \*, Carlos Domínguez-Ríos and Alfredo Aguilar-Elguézabal**

National Nanotechnology Laboratory, Advanced Materials Research Center S.C., Chihuahua 31136, Mexico; fernando.valenzuela@cimav.edu.mx (F.V.-D.l.R.); carlosdominguezrios560405@gmail.com (C.D.-R.); alfredo.aguilar@cimav.edu.mx (A.A.-E.)

\* Correspondence: roal.torres@cimav.edu.mx

**Abstract:** FeCrCB alloys have become an attractive option as a hardfacing coating to extend the service life of tools used in primary and secondary industries. In this work, experiments are reported on the sintering of FeCrCB alloy powders for hardfacing coatings by modifying the $CO_2$/CO ratio using six different atmospheric gas conditions. The hardfacing coating was found to have higher microhardness and higher abrasion resistance under a 10C atmosphere. This increase in mechanical properties is related to the microstructure, as the atmosphere using 10C promotes the formation of a higher quantity of hard phases, while the presence of CO induces the formation of higher volumetric fractions of eutectic phases, and, consequently, lower abrasion resistance is obtained.

**Keywords:** abrasion; slurry; sintering; tape casting; $CO_2$/CO atmosphere; hardfacing





## 1. Introduction

In the agricultural sector, abrasive wear is the leading cause of premature mechanical failure in soil removal tools. Farmers are affected by the continuous manual labor downtime and the cost of replacing worn-out parts. These worn-out tools are less effective in cultivating the land, which implies higher production costs as well as emissions penalties [1]. Wear has been defined in different ways, most of which involve a process of material loss in which two surfaces slide against each other. Traditionally, wear mechanisms are classified as adhesion, abrasion, erosion, fatigue and chemical wear [2]. Abrasive wear results in material loss due to the dynamic interaction of two surfaces against each other. The superficial removal of material from the surface results in dimensional losses. In a closed system, the lost material (material burr) causes an increase in the wear rate, which is achieved by the presence of the three bodies of abrasive wear, which is characterized by the relative motion of the abrasive particles over two phases in contact [3].

Multiple surface modification techniques, such as surface coatings and hardfacing, have been developed over the years, with the aim of mitigating the abrasive wear problem in agricultural implements [4–7]; however, for the particular case of tools used in soil removal, several researchers have stated the proven success of hardfacing coatings when the resistance exceeds very severe abrasive conditions, or when downtime becomes longer, and when it becomes apparent that hardfacing is cheaper than designing the entire part from an improved material [1].

Hardfacing is a metal forming technique in which a hard or resistant material is placed on the surface of a substrate made from another material. The hardfacing alloy is deposited uniformly on the surface of the base material by means of welding, so as to improve the hardness and abrasion resistance without modifying the ductility and toughness of the base material; it is also a flexible technique that allows the development of surfaces of different metals and alloys on a metallic base material, so that they can withstand wear as well as prevent corrosion and oxidation at high temperatures [8].

The state-of-the-art hardfacing alloys include low-cost FeCrC or FeCrB alloy systems and, on the other hand, high-cost multiphase composites containing combinations of tungsten carbide, niobium or titanium [9]. FeCrB castings have recently been developed to replace FeCrC-type iron castings, mainly due to the discovery of the solid solution strengthening effect of boron and its role in promoting the precipitation of carbides, resulting in carbide volume fraction (CVF) increases from 14.10 to 36.00% with the boron increasing from 0 to 1.4 wt.% in the alloy [10]. In the hardfacing industry, researchers have attempted to add boron to FeCrC-type hardfacing alloys to develop new FeCrCB-type hardfacing alloys [11].

During the hardfacing process, the primary purpose of the sintering atmosphere is to control chemical reactions between the alloy components and their surroundings. The second purpose is to remove the decomposition products of the used lubricants released during the preheating process. The importance of controlling the chemical reactions becomes evident when considering the high porosity contained in the green compacts. Thus, gases in the sintering atmosphere can not only react with the outer surface of the compacts, but can also penetrate the porous structure and react with the inner surfaces of the compacts. Methane, propane and other hydrocarbon gases can be partially combusted with air and obtain the combustion products $H_2O$, $H_2$, $N_2$, $CO_2$ and CO, as well as small amounts of unburned methane, when available, with a low air–gas ratio [12].

In the literature related to research on the influence of sintering atmospheres on Fe-based metal powders, there are several classifications of atmospheres; for example, Fridman [13] classified atmospheres as endothermic and exothermic, based on their $CO/CO_2$ ratio. In other research works [14–21], it is mentioned more generally that atmospheres can be classified into inert atmospheres ($N_2$, Ar and He gases) and reactive atmospheres. These are subdivided into oxidants and carburizers/decarburizers. Reactive atmospheres are generally composed of CO, $CO_2$, propane, endogas, $H_2O$ and $H_2$, among which those containing $H_2$, and, to some extent, CO and inert gases, are reductive, and these atmospheres are mainly used for sintering Fe-based compacts. Atmospheres containing a mixture of CO, propane, acetylene and endogas have a carburizing character; on the other hand, atmospheres with a high content of water vapor, $CO_2$, O and endogas are decarburizing atmospheres. The selection of the sintering atmosphere for Fe-based powder compacts must take into consideration the composition of the alloy to be sintered, the affinity of the alloying elements of the ferrous alloys and the final mechanical characteristics of the parts to be made by powder metallurgy.

The objective of this work was to study the effect of a carburizing–decarburizing sintering atmosphere on the abrasion resistance of a hardfacing coating on a boron steel substrate by applying the hardfacing alloy via the tape casting method with a suspension in water using water-atomized metal powders.

The atmospheres proposed in this study have a different $CO/CO_2$ ratio and were chosen to simplify the atmosphere created by the combustion of methane, propane and other hydrocarbon gases, which are the source of industrial heating in industrial furnaces. For the production of Fe-based alloy parts obtained by powder metallurgy, it is possible to form carbides with carbide-forming elements since they can contribute with C by virtue of the $CO/CO_2$ ratio; thus, the aim of this work was to study their influence on the microstructure of the hardfacing coating obtained and, consequently, on the mechanical performance.

## 2. Materials and Methods

### 2.1. Hardfacing Coating of Substrates

The choice of working conditions was made based on a previous work [22] that considered the results of the dilatometric test where the maximum contraction of the green tape cast was found to be around 1140 °C with a dwell time of 10 min, besides considering some other results [23]. The chemical composition of the FeCrCB hardfacing powders used in this research work can be observed, as well as the composition of the slurry being

applied by the tape casting method, in Tables 1 and 2. The chemical composition of the boron steel substrate is shown in Table 3.

**Table 1.** Chemical composition of the metallic powder.

| Element | B | C | Cr | Mn | Ni | Si | Fe |
|---|---|---|---|---|---|---|---|
| Weight percent (%) | 1.320 | 3.230 | 9.318 | 1.520 | 3.956 | 5.500 | Balance |

**Table 2.** Composition of the slurry used for tape casting.

| Compound | Weight Percent (%) |
|---|---|
| Metallic Powder | 89 |
| Flux | 2 |
| Deionized Water | 9 |

**Table 3.** Chemical composition of the steel substrate.

| Element | C | Si | Mn | P | S | Cr | Al | B | Fe |
|---|---|---|---|---|---|---|---|---|---|
| Weight percent (%) | 0.335 | 0.224 | 1.190 | 0.012 | 0.005 | 0.199 | 0.038 | 0.001 | Balance |

As shown in Table 4, the gas atmospheres used in this study involved different $CO/CO_2$ ratios.

**Table 4.** Hardfacing coating name, sintering atmosphere and $CO_2/CO$ rate in the sintering atmosphere.

| Atmosphere | CO$_2$/CO Rate | Hardfacing Coating |
|---|---|---|
| 100% $CO_2$ | 10 | 10C |
| 90% $CO_2$ + 10% CO | 9 | 9C |
| 70% $CO_2$ + 30% CO | 2.3 | 7C |
| 50% $CO_2$ + 50% CO | 1 | 5C |
| 30% $CO_2$ + 70% CO | 0.4 | 3C |
| 10% $CO_2$ + 90% CO | 0.1 | 1C |

The boron steel substrate was cut into samples of 25.4 mm × 76.2 mm × 6.35 mm. The pieces were sandblasted to remove the oxide layers, grease and dirt from the surface, before being placed in an ultrasonic bath with acetone for 5 min to remove any residual grease.

The slurry of metal powders, water and flux was vigorously stirred for 2 min or until a homogeneous slurry was formed, and then the homogeneous suspension was applied to the substrate using the instrument described in a previous work [22]. This implementation allows the application of a uniform layer of 2 mm thickness on the substrate in a single step. Thereafter, the substrate with the green coating was placed in an oven at 200 °C for 80 min to completely remove moisture.

The dried samples were placed in a furnace for the sintering process, using six different atmospheres (see Table 4). The heating conditions were the same for all atmospheres, 1140 °C for 10 min, with a heating rate of 23 °C/min. At the end of the 10 min period, the samples were removed from the oven and rapidly cooled in water at room temperature. For X-ray diffraction, the samples were cut into pieces of 25.4 mm × 25.4 mm × 8 mm. X-ray diffraction was carried out on the surface of the coating, having the necessary thickness so as to avoid interference by the substrate. For samples used for optical microscopy and scanning electron microscopy analyses, the cross-sectional area was observed and these same samples were used for microhardness testing. For the abrasion resistance test, the ASTM G65 standard suggests samples of 25.4 mm × 76.2 mm with a thickness not greater than 12.7 mm, and it also suggests performing three tests for each condition; we used procedure B [24] in this work (see Table 5).

**Table 5.** Abrasion resistance test conditions according to ASTM G65 standard.

| Specific Procedure | Force against Specimen (N) | Wheel revolutions (rpm) | Linear Distance (m) | Speed (rpm) | Sand Flow (gmin$^{-1}$) |
|---|---|---|---|---|---|
| B | 130 | 2000 | 1436 | 200 +/− 10 | 300–400 |

### 2.2. X-ray Diffraction Analysis

The 25.4 mm × 25.4 mm × 8 mm samples were analyzed in a PANalytical X'Pert Plus brand X-ray diffractometer (Malvern Panalytical Ltd., Malvern, UK) under the following conditions: step size of 0.3°, step time of 10 s and range from 20° to 80°.

### 2.3. Microhardness Characterization

The microhardness of the sintered coating cross-sections was evaluated using a Future Tech MH-00 microdurometer (Future-Tech Corp, Kawasaki, Japan). Two different metallographic conditions were used, where the first measurement condition was carried out on the polished surface with a load of 500 g and a loading time of 12 s; 10 measurements were taken on the polished surface with ten probes for every hardfacing coating, and the second condition involved the chemical etching of the surface to reveal the microstructure; 10 measurements were taken in the hard phases and 10 measurements in the eutectic phase, using a load of 50 g and a loading time of 10 s. The applied load changed due to the size of the phases present, which prevented the indentation traces from being positioned in both phases at the same time.

### 2.4. Abrasion Resistance

The abrasion resistance was evaluated using procedure B of the ASTM G65 standard [24], with the test conditions shown in Table 5; this was chosen to simulate the abrasive wear of agricultural implements under normal conditions [25]. Three tests were performed on each sample, as suggested by the standard method, using abrasive particles of Ottawa silica sand, as presented in Figure 1. This test yielded three results, volume loss, wear rate and specific wear rate, which were calculated using Equations (1)–(3), respectively.

$$Volume\ loss\ \left(mm^3\right) = \frac{weight\ loss\ (g)}{density\ \left(\frac{g}{cm^3}\right)} * 1000 \left(\frac{mm^3}{cm^3}\right) \tag{1}$$

$$wear\ rate = \frac{volume\ loss\ (mm^3)}{sliding\ distance\ (m)} \tag{2}$$

$$specific\ wear\ rate = \frac{volume\ loss\ (mm^3)}{sliding\ distance\ (m) * force\ against\ specimen\ (N)} \tag{3}$$

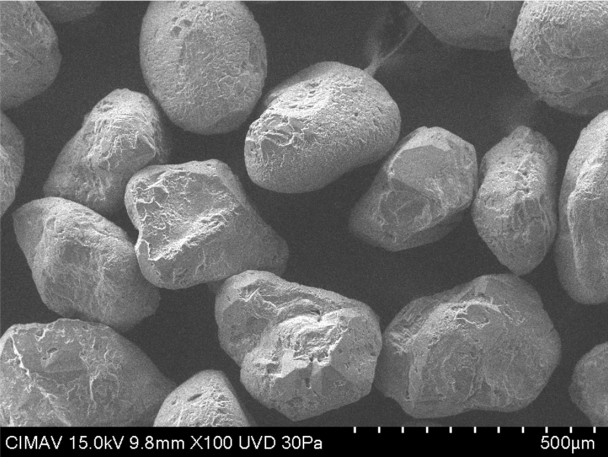

**Figure 1.** Ottawa silica sand used as abrasive particles in the abrasion resistance test.

## 3. Results and Discussion

### 3.1. X-ray Diffraction

The diffraction patterns of the metal powders as received from the supplier and the furnace-sintered hardfacing coatings with six different atmospheres are presented in Figure 2. The phases present in the initial metallic powders received were solid solutions: $Cr_{0.1}Fe_{0.63}Si_{0.27}$ and FeCr—the carbides present were $Cr_7C_3$ and $Fe_3C$, and the borides were $Fe_2B$ and $Mn_2B$. After the sintering process, the structure evolved and the solid solutions dissolved, which left only $Fe_{0.87}Cr_{1.13}$, with the carbides increasing to $Fe_3C$, $Fe_7C_3$ and $Mn_7C_3$, and the borides increasing to $Mn_2B$, $CrB_2$, $Fe_2B$ and $Fe_3B$. The carbide and boride phases have been reported as $M_7C_3$, $M_{23}C_6$, $M_2B$ and $MB_2$, where M can be Fe, Cr, Mn or any combination of these [11,26,27].

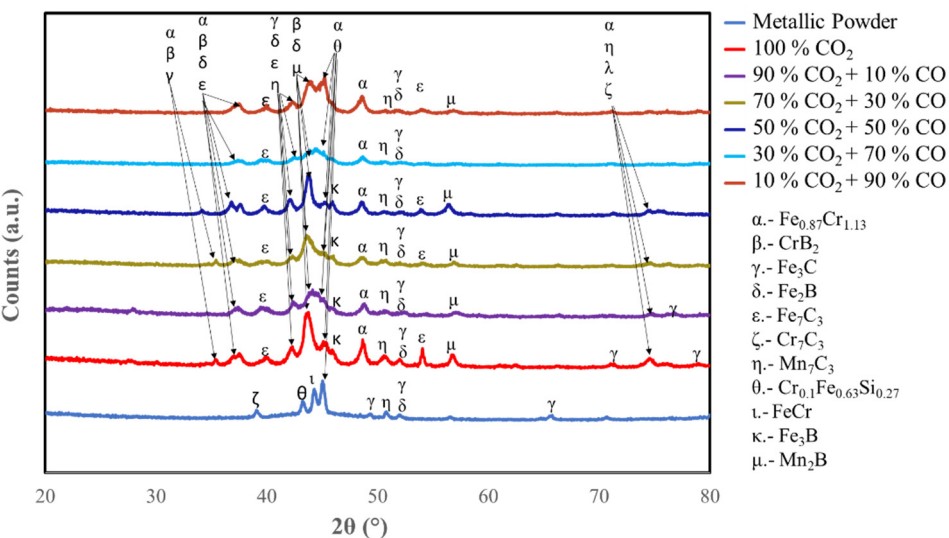

**Figure 2.** XRD of the metallic powder as received and the hardfacing sintering in an oven with six different atmospheres.

### 3.2. Optical Microscopy

Samples of 17.4 mm × 10 mm × 8 mm were cut from the furnace-sintered hardfacing coatings using six different sintering atmospheres (see Table 4), and the cross-section was analyzed to determine if there were any microstructural differences caused by the varying conditions of the sintering atmospheres. Figure 3 shows micrographs of the furnace-sintered samples under six different atmospheres, and it can be seen that there is apparently no significant difference in the phases present, regardless of the atmosphere used during the sintering process. On the other hand, it can be seen that the hard phases were surrounded by laminar eutectic phases, such as those reported by Berns and Fisher [26]. Table 6 shows the volumetric fractions of the three phases present in the studied samples, namely the hard phase; the hard phases were constituted by different primary carbides, $M_7C_3$ and/or MC, where M could be Fe, Cr, Mn or any combination of these, and the eutectic phases were constituted by $M_{23}C_6$, $M_2B$ and $MB_2$ and a vitreous phase.

Table 6 shows that the atmosphere with the greatest production of hard phases (formed mainly by primary carbides $M_7C_3$ and/or MC) was 10C, while the atmosphere with the least production was 1C, and this same atmosphere increased the volumetric fraction of the vitreous phase. The atmosphere that produced the greatest amount of eutectic phases was 5C, and the relationship between microstructure and abrasion resistance has been reported by Hryhaa et al. [19]. Namely, increasing the volumetric fraction of hard phases increases the abrasion resistance.

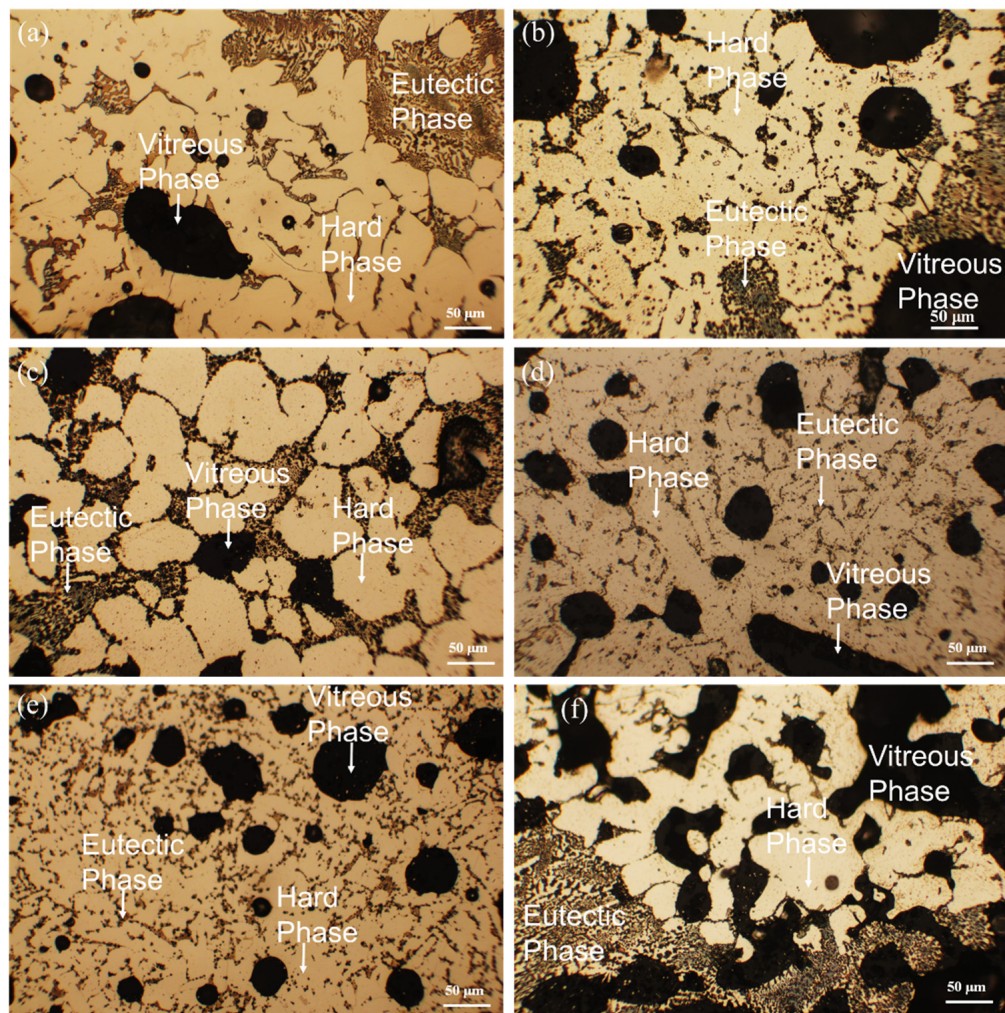

**Figure 3.** Micrographs of the cross-sectional hardfacing sintering in an oven with different atmospheres: (**a**) 10C, (**b**) 9C, (**c**) 7C, (**d**) 5C, (**e**) 3C and (**f**) 1C.

**Table 6.** Volumetric fractions of the phases presented in six different hardfacing coatings.

| Hardfacing Coating Name | Volume Fraction Phase | | |
|---|---|---|---|
| | Vitreous | Eutectic | Hard |
| 10C | 0.15 | 0.16 | 0.69 |
| 9C | 0.2 | 0.18 | 0.62 |
| 7C | 0.19 | 0.22 | 0.59 |
| 5C | 0.14 | 0.4 | 0.46 |
| 3C | 0.17 | 0.24 | 0.59 |
| 1C | 0.32 | 0.22 | 0.45 |

As observed in Figure 4, the behavior of the fraction of phases conforms to the microstructure of the sintered hardfacing coatings under different atmospheres. It can be seen that when the $CO_2/CO$ ratio is 1 (5C), the quantity of hard phases is lower (Figure 4a), which is related to a smaller quantity of primary carbides. It can also be observed that, under this atmospheric composition, a greater quantity of eutectic phases is obtained (Figure 4b). This figure also shows that with increasing amounts of $CO_2$ in the sintering atmosphere, a larger volume fraction of hard phases is formed, most likely because this atmosphere allows the introduction of interstitial elements, such as carbon.

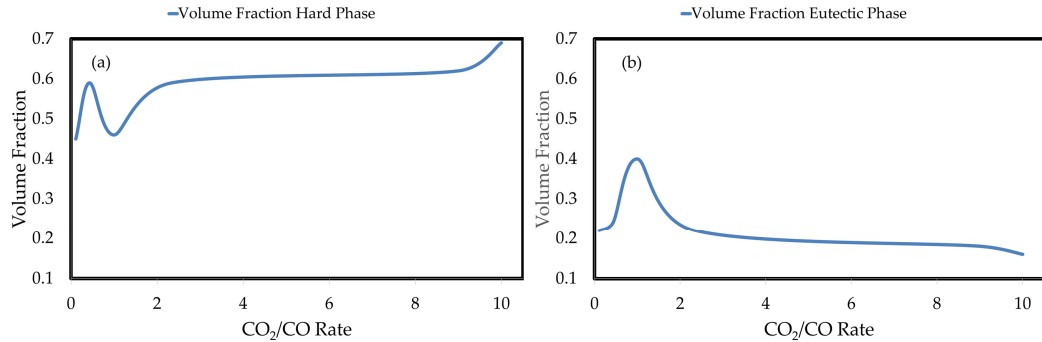

**Figure 4.** (**a**) Volumetric fraction of hard phase and (**b**) volumetric fraction of eutectic phase, vs. $CO_2/CO$.

### 3.3. Scanning Electron Microscopy

A cross-section of the sintered hardfacing coatings was analyzed using a Hitachi model SU3500 scanning electron microscope (Hitachi High-Tech in America, Schaumburg, IL, USA). Figure 5 shows the hard phases surrounded by a laminar-type eutectic phase, such as those described by Phillips et al. [15]. It can also be distinguished that in the 3C atmosphere, the eutectic phase cannot be seen in the photomicrograph shown due to the low amount present in the microstructure.

### 3.4. Microhardness

Table 7 shows the microhardness results for the eutectic phase, hard phase and un-etched samples. Figure 6 shows graphically the average values of ten determinations. The results show that the hardness values of the hard phase tend to be high when the CO content in the sintering atmosphere increases. On the other hand, when an inert atmosphere (Ar gas) was used, as in the previous study [22], microhardness values of 832.5 HV and 958.9 HV were reported for samples cooled in air and water at room temperature, respectively; it can be observed that the microhardness values in $CO_2/CO$ atmospheres are higher than those previously reported [22]. As can be observed, in general, the hardness of the hardfacing coating increases when $CO_2/CO$ atmospheres are used; the increase in hardness may be attributed to the increase in carbide and boride formation.

**Table 7.** Average Vickers hardness values of the sintered samples.

| Hardfacing Coating | Microhardness (HV) | | |
|---|---|---|---|
| | Eutectic Phase | Hard Phase | Unetched Samples |
| 10C | 887.2 +/− 146 | 1130.7 +/− 130 | 988.7 +/− 147 |
| 9C | 843.7 +/− 80 | 1147.7 +/− 130 | 845.5 +/− 141 |
| 7C | 843.5 +/− 98 | 1120.7 +/− 88 | 1013.8 +/− 166 |
| 5C | 786.3 +/− 82 | 1215.6 +/− 76 | 986.5 +/− 134 |
| 3C | 826.5 +/− 139 | 1230.1 +/− 111 | 1040.5 +/− 62 |
| 1C | 880.1 +/− 79 | 1198.6 +/− 141 | 999.4 +/− 160 |

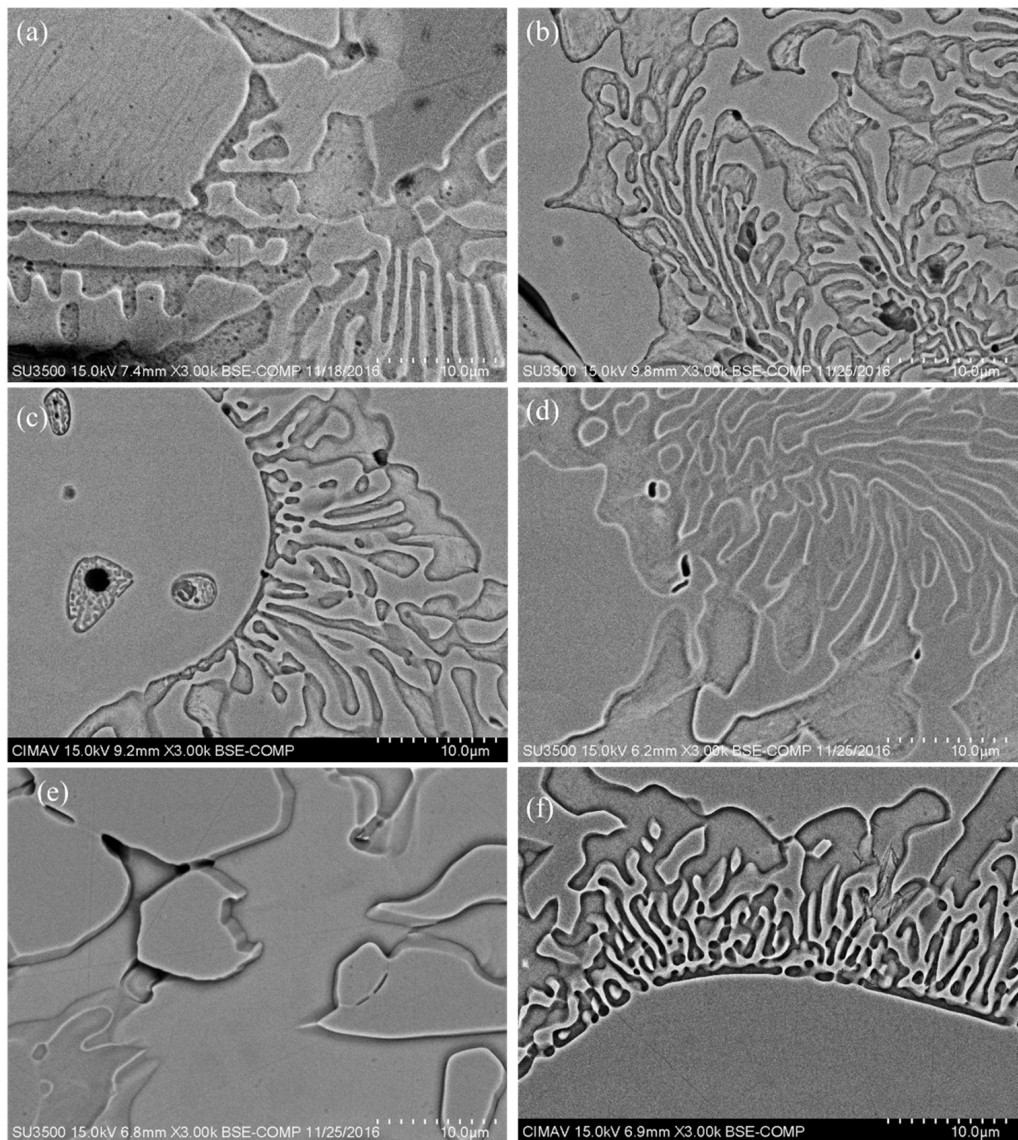

**Figure 5.** SEM cross-sectional micrographs of the hardfacing coatings sintered in six different atmospheres: (**a**) 10C, (**b**) 9C, (**c**) 7C, (**d**) 5C, (**e**) 3C and (**f**) 1C.

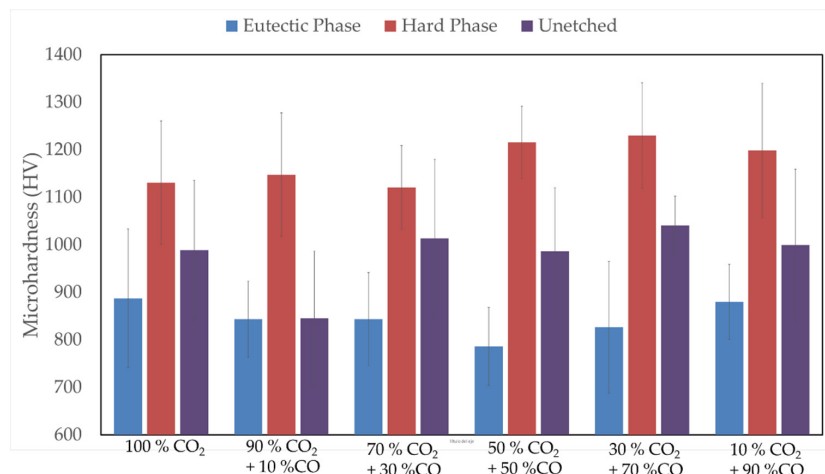

**Figure 6.** Microhardness of the hard phase, eutectic phase and unetched hardfacing coatings sintered under six different atmospheres.

### 3.5. Abrasion Resistance

The results of the abrasion resistance test are presented in Table 8 in terms of volume loss, wear rate and specific wear rate. The microhardness of the Ottawa silica sand is also presented. The average hardness of the hardfacing coatings sintered in six different atmospheres (see Table 7), and the Ha/Hc ratio, which corresponds to the hardness of the abrasive particles divided by the hardness of the coating, have an important relationship because, if its value is between 0.7 and 1.1, there is no abrasive wear; however, if this relationship is between 1.3 and 1.7, it indicates the occurrence of maximum wear [28–31].

**Table 8.** Results of wear tests according to the international ASTM G65 standard for hardfacing alloy sintered under different atmospheres.

| Hardfacing Coating | Microhardness of Sand (HV) | Microhardness (HV) | Ha/Hc | Volume Loss ($mm^3$) | Wear Rate ($mm^3 m^{-1}$) | Specific Wear Rate ($1 \times 10^{-14}$ $m^2 N^{-1}$) |
|---|---|---|---|---|---|---|
| 10C | | 988.73 | 1.32 | 14.26 +/− 0.19 | 0.01 | 7.64 |
| 9C | | 845.48 | 1.54 | 15.32 +/− 0.19 | 0.01 | 8.21 |
| 7C | | 1013.75 | 1.29 | 19.02 +/− 3.74 | 0.01 | 10.18 |
| 5C | 1304.96 | 986.48 | 1.32 | 28.22 +/− 2.45 | 0.02 | 15.12 |
| 3C | | 1040.53 | 1.25 | 23.42 +/− 0.19 | 0.02 | 12.55 |
| 1C | | 999.43 | 1.31 | 21.27 +/− 0.19 | 0.01 | 11.39 |

In Figure 7, the results for abrasion resistance and microhardness are shown as a function of the volume fraction of the phases present in the hardfacing coating after sintering, and it can be seen that the greater the hard phase, the higher the abrasion resistance, which is achieved by sintering in a 100% $CO_2$ atmosphere. The hardfacing coatings sintered in an atmosphere of 50% $CO_2$ + 50% CO and the hardfacing coatings sintered in an atmosphere of 10% $CO_2$ + 90% CO had almost the same volume fraction of the hard phase, while the abrasion resistance of the hardfacing coatings was different, with the first one being higher. This behavior can be attributed to the amount of eutectic phases present; therefore, to obtain the best abrasion resistance, the amount of hard phases must be maximized and the amount of eutectic phases must be minimized. Furthermore, no vitreous effects on the phases were observed in the hardfacing coating performance.

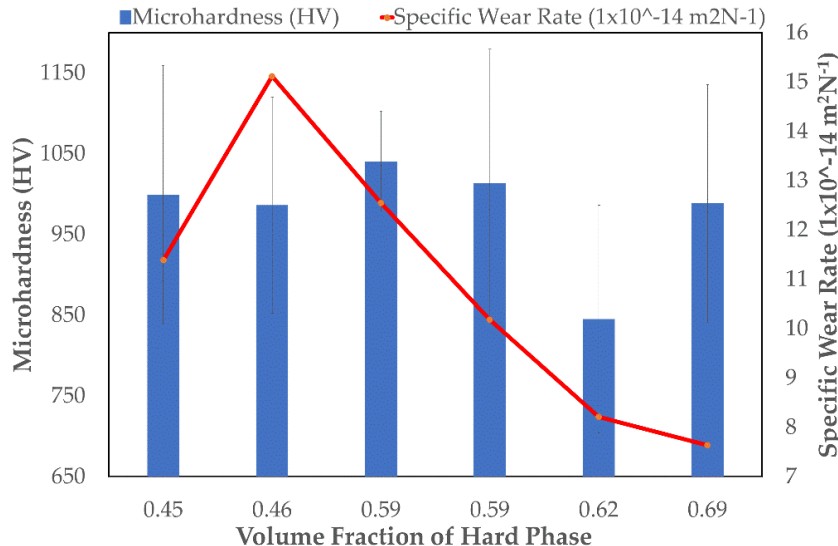

**Figure 7.** The relation between volume fraction of hard phase, microhardness and specific wear rate.

Micrographs of the worn area of the sintered hardfacing coatings are shown in Figure 8. As can be seen in Figure 8a, discontinuous lines are observed, which are the product of plastic deformation produced by the micro-plowing mechanism, in addition to the presence of holes observed, which are characteristic of the abrasion of the three bodies. For the sample in Figure 8b, there are continuous lines (grooves) produced by micro-plowing and

the presence of metallic burrs is also observed, which is produced by the detachment of particles from the hardfacing coating, while the samples in Figure 8c,d show discontinuous lines similar to those observed in Figure 8a, and the amount of metallic burrs is higher than that observed in Figure 8b. For the samples produced under atmospheres with higher CO content, i.e., Figure 8e,f, continuous lines are observed with a higher amount of metallic burrs, and the wear mechanisms presented in the six studied atmospheres are micro-plowing and three-body wear, which promote the formation of metallic burrs; however, there is an evolution of the micro-plowing main wear mechanism towards three-body wear by increasing the amount of CO in the sintering atmosphere, which is in agreement with findings presented by other researchers [29,32–34].

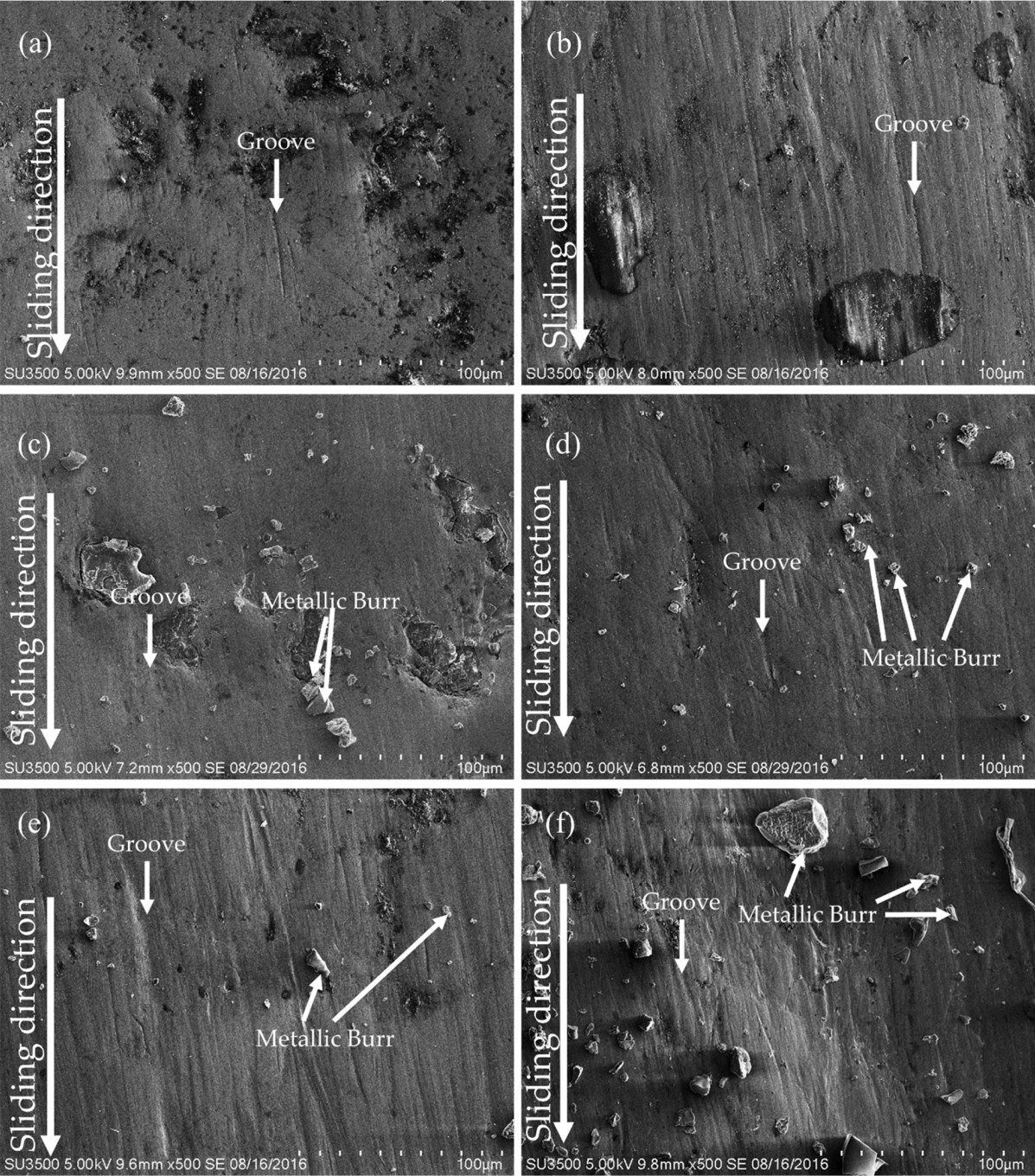

**Figure 8.** Micrographs of the worn-out surfaces after the ASTM G65 abrasion resistance test for hardfacing coatings sintered in an oven under six different atmospheres: (**a**) 10C, (**b**) 9C, (**c**) 7C, (**d**) 5C, (**e**) 3C and (**f**) 1C.

From a structural point of view and from a tool life perspective regarding the use of these hardfacing coatings, the predominance of the micro-plowing mechanism suggests a longer useful life of the tools than when three-body wear is used as the main wear mechanism, due to the fact that the former includes a plastic deformation step before failure. This step is associated with energy absorption, while the three-body mechanism produces the formation of metallic burrs with minimal plastic deformation, which facilitates material removal and therefore a shorter tool life.

## 4. Conclusions

The metallic powders used to develop the hardfacing coatings had six main types of phases, which were $Cr_{0.1}Fe_{0.63}Si_{0.27}$, $FeCr$, $Cr_7C_3$, $Fe_3C$, $Fe_2B$ and $Mn_2B$. After the sintering process, these phases evolved to $Fe_{0.87}Cr_{1.13}$, $Cr_7C_3$, $Fe_3C$, $Fe_2B$ and $Mn_2B$.

The microstructure obtained by the process of applying hardfacing coatings consisted of three well-defined phases, which are a hard phase, surrounded by a laminar eutectic and a vitreous phase (hard phases are identified with carbides or borides)

The volumetric fraction of the phases varied based on the sintering atmosphere used; thus, the highest content of the hard phase was reached with an atmosphere of 10C (0.69), while the least content was obtained with an atmosphere of 1C (0.45).

There was a significant difference in the microhardness of the hard and eutectic phases, having a minimum value in the 10C (243.5 HV) atmosphere and a maximum difference in the 5C (429.3 HV) atmosphere, with these differences affecting the average microhardness of the hardfacing coatings and increasing the standard deviation.

There was a relationship between microhardness and resistance; however, this relationship was not direct because the atmosphere (3C) that produced the highest microhardness (1040.5 HV) did not have the best abrasion resistance (23.4 mm$^3$), and the sample with the best abrasion resistance (14.3 mm$^3$) was obtained with a hardness of 988.7 HV using a 10C atmosphere.

The relationship between abrasion resistance and microstructure became evident when we observed that the best abrasion resistance (14.3 mm$^3$) was achieved while having the highest amount of the hard phase (0.46), and the worst abrasion resistance (28.2 mm$^3$) was obtained while having the highest amount of the eutectic phase (0.40).

The main wear mechanisms of sintered hardfacing alloys in the six different atmospheres (see Table 4) were micro-plowing and three-body wear, where the first presented itself as lines formed along the path of the abrasive particles for plastic deformation, while the latter produced metallic burrs. Micro-plowing was the main mechanism for higher concentrations of $CO_2$ in the atmosphere of the hardfacing treatment, and by increasing the CO content of the atmosphere, the main wear mechanism became three-body wear.

Regarding the conditions evaluated in this work, it is preferable to have a micro-plowing mechanism before failure due to the energy absorption produced during plastic deformation, while the wear of the three bodies does not present this energy absorption before failure due to their minimal plastic deformation before the formation of the metallic burrs.

**Author Contributions:** Conceptualization: F.V.-D.l.R. and C.D.-R.; methodology: F.V.-D.l.R., C.D.-R., R.T.-S. and A.A.-E.; formal analysis: F.V.-D.l.R., C.D.-R., R.T.-S. and A.A.-E.; investigation: F.V.-D.l.R. and C.D.-R.; writing—original draft preparation: F.V.-D.l.R. and C.D.-R.; writing—review and editing: F.V.-D.l.R., C.D.-R. and A.A.-E.; investigation and visualization: F.V.-D.l.R.; supervision: C.D.-R. All authors have read and agreed to the published version of the manuscript.

**Funding:** This research received no external funding.

**Institutional Review Board Statement:** Not applicable.

**Informed Consent Statement:** Not applicable.

**Acknowledgments:** We acknowledge César C. Leyva Porras and Karla Campos Venegas for their help with scanning electron microscopy, and José T. Holguín Momaca for his help with X-ray diffraction.

**Conflicts of Interest:** The authors declare no conflict of interest.

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
