# Peer review of "Characterization of the Effect of the CO2/CO Sintering Atmosphere on the Abrasion Resistance of a FeCrCB Hardfacing Coating"

_metals, doi:10.3390/met11101568_

Round 1
Reviewer 1 Report
Dear authors,
the paper is interesting but it could be improved. There are some suggestions:
- Wear tests conditions (lines 207 - 213), Table 4 and Fig. 6 should be relocated to Methods section.
- Section 2.1: Please add tables with compositions of the substrate steel, coating, and slurry.
- Line 155 "hard, eutectic and vitreous phases": Please mark these phases on Fig. 2. It would be very good to discuss their compositions somewhere in the paper. Could you make also some suggestions about reactions stipulated by various CO2/CO ratios?
- Line 49 "strengthening effect of boron and its role in promoting the precipitation of carbides": I could suppose that B in Me1-Me2-B is forming borides. Please clarify that paper [10] describes processes in Fe-Cr-C-B systems with 2 wt% of carbon.
- Please correct sections "Authors contribution" and "Acknowledgements"
- Please check the style through the paper and correct some misprints:
Line 92 "some other works": probably it could be replaced by "some other results"
Line 102 "anººd" Correct it please.
Line 149 "the atmosphere sintering": I suppose it should be "sintering atmosphere"
Reviewer 2 Report
The paper ”Characterization of the effect of the CO2/CO sintering atmosphere on the wear resistance of a FeCrCB hardfacing coating” can be published also in Metals, but can be published in Coatings Journal. The paper consists of significant results in order to be accepted.
Author Response
Thank you.
Reviewer 3 Report
See the attachment.

Round 2
Reviewer 3 Report
Generally, the manuscript was improved, a few comments on the revised version:
1. The current Figure 1 does not really needed in the main manuscript, the authors can put it in an SI file.
2. The presentation of Tables are not well organized, they look like in a random style, especially Table 5.
3. The phase identification in Figure 2 is not convincing, the signals are too weak to identify some phases with only 1 or 2 peaks in the measured XRD patterns.
4. The quality or resoloution of Figures 4 & 6 needs to be improved. They do not look like original images.
5. Still think the conclusions in the last 2 paragraphs, especially the last paragraph, have no strong relation to the contents in the main text.
6. There are still errors, such as "DRX" in the caption of Figure 2, it should be "XRD". Places like this must be changed.
Author Response
Please se attachment
